# Design of Plasmonic-Waveguiding Structures for Sensor Applications

**DOI:** 10.3390/nano9091227

**Published:** 2019-08-29

**Authors:** Jaroslav Vlček, Jaromír Pištora, Michal Lesňák

**Affiliations:** 1Nanotechnology Centre, VŠB—Technical University of Ostrava, 708 00 Ostrava, Czech Republic; 2Department of Mathematics and Descriptive Geometry, Faculty of Mechanical Engineering, VŠB—Technical University of Ostrava, 708 00 Ostrava, Czech Republic; 3IT4Innovations, VŠB—Technical University of Ostrava, 708 00 Ostrava, Czech Republic

**Keywords:** surface polaritons, planar waveguide, mode coupling, optical sensors

## Abstract

Surface plasmon resonance has become a widely accepted optical technique for studying biological and chemical interactions. Among others, detecting small changes in analyte concentration in complex solutions remains challenging, e.g., because of the need of distinguishing the interaction of interest from other effects. In our model study, the resolution ability of plasmonic sensing element was enhanced by two ways. Besides an implementation of metal-insulator-metal (MIM) plasmonic nanostructure, we suggest concatenation with waveguiding substructure to achieve mutual coupling of surface plasmon polariton (SPP) with an optical waveguiding mode. The dependence of coupling conditions on the multilayer parameters was analyzed to obtain optimal field intensity enhancement.

## 1. Introduction

The coupling of resonance states in various optical nanostructures offers many promising ways for the development of new photonic devices. This expectation follows the recent results obtained for plasmonic coupled modes occurring in plasmonic metal-dielectric nanostructures [1,2]. The coupling between surface plasmon polariton and waveguide mode belongs to the intensively investigated effects, especially as the optical Fano effect (OFE) [3,4].

Surface plasmon resonance (SPR) has become a widely accepted optical technique for studying biological and chemical interactions. Among others, detecting small changes in analyte concentration in complex solutions remains challenging, e.g., because of the need of distinguishing the interaction of interest from other effects. In particular, non-specific binding and/or background refractive index changes are problematic in the multi-component analytes, e.g., in medicine, food safety and environmental applications. Typically, the SPR sensor is characterized by its sensitivity, penetration depth, and perhaps even by full width of half-minimum (FWHM) of the reflectance dip. In the angular interrogation mode, the traditional sensing device has an angular sensitivity of 50–100 deg/RIU, where RIU denotes refractive index units. If the reflectance shift ΔR at fixed incidence angle is detected instead SPR dip angular shift, the sensitivity moves about 30 RIU^−1^ [2]. Depending on applied measuring method the resolution limit less than 10^−4^ can be achieved [5]. Penetration depth gives quite reliable indication of the usable distance, at which the SPR device is sensitive to changes in the analyte (≈200 nm at optical frequencies [6]).

To improve sensing parameters, several different ways were suggested to enhance detection limit of SPR sensors, penetration depth of those or specific application conditions:adding of photonic crystal predetermining effective refractive index that supported SPP modes excitation [7,8,9];implementation of insulator-metal-insulator (IMI) or metal-insulator-metal (MIM) substructures that produce long-range surface plasmons (LR SPP) with lower absorption losses [6,10];coupling of SPP and waveguide (WG) modes (see, e.g., [3,11]);dispersion plasmonic interaction at an interface between a doped semiconductor and a dielectric [12];applications of new materials (e.g., graphene) and/or specific conditions as cryonic arrangement at IR frequencies [13,14]; andsome combination of the above items.

In our recent work, we analyzed the high refractive index waveguide based on ferromagnetic garnet [11]. Some results obtained there with relation to the magneto-plasmonic sensing showed certain perspective by mutual coupling of the resonant states excited in parallel in plasmonic and waveguiding sub-system. Supplementing two or more metal-dielectric interfaces close to each other into fundamental waveguide brings interesting effects: besides the coupling between SPPs of individual interfaces, the other kinds of combined resonance states are generated dependently on the material and/or geometrical properties of separate components.

This paper is devoted to the coupling of resonance states arising in optical structure combining planar waveguide (WG) with MIM SPR system, when the both sub-structures operate in arrangement with the same coupling prism. In the next section, the basic model approach is described as follows. Planar waveguide parameters as the background of waveguide-plasmonic effects are introduced in Section 2.1, with special attention paid to the long- and short-range plasmons excited in the MIM complement that is analyzed in Section 2.2. Several forms of the coupling waveguide modes with plasmonic ones are presented in Section 2.3, and resulting combined resonance states are discussed regarding their sensing ability in Section 2.4. In Section 3, the SPP-WG sensor setup is proposed including sensing ability expressed by achievable sensitivity and resolution limit.

Numerical results presented here were obtained using 4 × 4 matrix method based on rigorous coupled-waves algorithm (RCWA, [15]) implemented as its own Matlab code. The presumed experimental arrangement (detection part) for sensor response analysis is based on lock-in detection technique. In the first step of structure testing, tens of kHz operating frequency range will be applied.

## 2. Methods

### 2.1. Waveguiding Structure

The bismuth-doped gadolinium iron garnet waveguide layer of thickness *d* (Bi:GIG, $n_{2}=2.4619-0.0042i$) was prepared on a gallium-gadolinium garnet substrate (GGG, $n_{3}=1.9648$) with refractive indices at the wavelength 633 nm [16]. The proposed structure was separated from the coupling prism by air-gap superstrate. As the coupling prism must have a material with a relatively high refractive index, the rutile prism (n=2.5836) [17] was used. This structure enabled generating standard TE and TM guided modes, where guided order was controlled by the Bi:GIG layer thickness (see Figure 1).

In the studied plasmonic-waveguiding structure, the GGG substrate was naturally finite; the water as adjoining analyte (n=1.332) was considered. The substrate thickness of 100 nm was fixed in all simulations because of its small influence on the studied effects compared with the other geometrical parameters of structure. Table 1 illustrates penetration depth into the GGG for the first three mode orders for the both fundamental polarizations of incident field.

Since the air gap separating mentioned sub-structures is not sufficient to obtain mutual coupling, embedding of an appropriate interlayer is needed, the material and thickness of which were specified in our previous simulations [11]. The waveguide coupling forces depend primarily on the air gap thickness; nevertheless, inserting another layer between the prism and WG changed coupling conditions differently for the various modes, as demonstrated in Figure 2.

In the previous work [11], the magneto-optical activity of ferromagnetic garnet was applied as the principal effect. Therefore, only one gold layer with the Al-doped zinc oxide (n=1.8) interlayer was used. For the SPP-PWG system discussed here, an interlayer with lower refractive index would be advantageous in the proposed metal-insulator-metal structure (see Section 2.2).

Placing the glass nanolayer (n=1.5) importantly changes the coupling strength between prism and waveguide (Figure 2). The incidence angle ϕ is related to the prism base/structure interface; in particular, to the interface prism base/interlayer in Figure 2. Note that the layer thicknesses in presented schemes do not correspond to the real situation.

Inserting gold film of the appropriate thickness below the waveguide layer causes angular shift of TE resonance minima. It enables, among others, to achieve the generation of both TE and TM modes at the same incidence angle, facilitating an important group of experiments. Figure 3 shows one from resulting waveguide modification.

### 2.2. Metal-Insulator-Metal Nanostructure

Consider a thin dielectric film of the thickness *t* sandwiched between two finite (but quite large) metal layers of the same refractive index. Generally, the dispersion relation of this symmetric MIM structure gives four resonance states that represent two plasmonic and two photonic modes [19], a particular setup of which depends on the material as well as geometrical parameters of SPR structure [2,10].

In the configuration discussed here with the rutile prism and SiO_2_ gap between gold layers, the photonic (i.e., guided) modes cannot exist. On the other hand, one SPP mode survives for all values *t* of gap thickness. This mode exhibits odd symmetry of longitudinal electric field component $E_{y}$, and, consequently even symmetry of transversal component $E_{z}$ (normal to the interface) (see Figure 4a,b).

For sufficiently thick dielectric interlayer, the other SPP type is also supported, having the opposite characteristics of the corresponding electric field components (Figure 4c,d). In agreement with described properties, these SPPs are termed asymmetric and symmetric, respectively. In both cases, the longitudinal components are continuous regarding the boundary conditions.

Note that frequently used nomenclature works with the long-range (LR) SPP in the case of asymmetric longitudinal component, and short-range (SR) SPP in the opposite situation. This notation is typical for IMI plasmonic structures, where propagation length of the symmetric SPP is expressively less than that of the asymmetric one [20,21]. For clarity, we also keep the LR/SR notation in this paper.

The thicknesses of lossy metallic layers must be sufficiently small in real situations. Therefore, 44-nm gold film at the wavelength 633 nm in applied Kretschmann configuration was used. This is the reason that mentioned common Au sublayer thickness is preserved in the designed structure. The size of single MIM parts importantly influences the parameters of plasmonic resonance states that brings an advantage by SPP-WG coupling. Figure 5 illustrates excitation of the both kinds of SPPs for two states with symmetric geometry but of different dielectric slab, and the asymmetric case, when the first gold layer thickness predominates. LR SPPs excited close to 25 deg exhibit small dependence on MIM geometry, whereas the SR SPPs are hardly modified.

### 2.3. SPP–PWG Coupling

Consider the example when the dielectric waveguide interlayer in Figure 2 (green component below the prism) is supplied by the symmetric MIM plasmonic structure discussed in the previous subsection. In the combination with water analyte, we obtain the resonance response in Figure 6. The TE waveguide modes are practically suppressedk the weakly coupled mode TM_0_ is outside the figure. However, the waveguide coupling of TM_1_ mode is enhanced by the narrow SR plasmonic mode.

The coupling between plasmonic and waveguiding modes can be reached using several methods following the SPP type and WG mode order as well. Besides changes of waveguide conditions (gap and/or waveguide layer thickness), the geometrical parameters of MIM sub-structure enable efficient tuning of reflectance response.

Arising of SR SPP dips at higher incidence angles (Figure 5) leads to the coupling with the low order waveguide modes similar to in the previous example. This is easily realized by the thickness asymmetry in MIM components of the form $10/t_{i}/34$, where $t_{i}$ is the dielectric interlayer thickness taken as user-specific parameter. In Figure 7, the SR SPP coupling with the TM_1_ waveguide mode is modeled through small changes of the thickness $t_{i}$.

We observe the enhancing of TM_1_ coupling forces caused by the witness of plasmonic resonance state. An increase of the interlayer thickness leads to the angular shift of SPP dip that successively goes across the WG resonance minimum. This phenomena is usually referred as the optical Fano effect (OFE) (see, e.g., [3] and references therein).

On the contrary, the coupling forces of TM_3_ mode predominate compared to the LR SPP mode (see Figure 6). Since the mutual coupling demands an alignment of both resonance states, we attenuate the WG coupling forces by enlargement of air gap up to 300 nm together with the application of the opposite MIM asymmetry comparing with the previous case. The resulting reflectance response in Figure 8 is again of the OFE type.

A fine tuning of the SPP-WG coupling is again realized through the dielectric interlayer thickness. Similarly, in this case, we can speak about the OFE.

### 2.4. Sensing Ability of Plasmonic-Waveguiding System

Consider the SPP-WG sensing device as in Figure 6 operating in the angular interrogation mode with TM polarized optical beam. The reflectance increment $\Delta R_{p}$ at a reference incidence angle is detected as the response on small changes of analyte refractive index $\Delta n_{a}$. Thus, the sensitivity *S* and resolution limit min($\Delta n_{a}$) are expressed as
(1)$$S=\frac{\Delta R_{p}}{\Delta n_{a}}\;\left[{\mathrm{RIU}}^{-1}\right],\;\min\left(\Delta n_{a}\right)=\frac{\delta R_{p}}{S}\;\left[\mathrm{RIU}\right],$$
where $\delta R_{p}$ is the minimal limit for a given setup. The above findings allow designing of the sensoric structure with optimal operating parameters. At first, we exploit the enhancing of coupling forces by LR SPP coupling with a WG mode of high order that implies sufficient penetration depth of the probe field into the analyte. Choosing an appropriate reference incidence angle is the other important step. Waveguide resonance dips are not advantageous to the modulation because of their steepness, and too weak variability relating to a change of analyte refractive index. However, when the LR SPP dip is “locked” onto waveguide one, the resulting optical Fano effect not only produces enhanced electromagnetic field across the structure but also leads to span of WG dip. Thus, the resulting coupled resonance state disposes of required sensing properties.

## 3. Results and Discussion

We showed that the mutual coupling of SPR modes excited in MIM structure with guided modes in linked waveguiding sub-system allows establishing various coupled resonance states formatting favorable conditions for sensoric applications. The 140 nm dielectric strip together with the 170 nm air gap led to quite close comparable reflection dips that create sensitivity increase of coupled resonance state (Figure 9 and Figure 10).

The segment of reflectance curve between green points in Figure 9b is the most efficient to the sensitivity testing because of its linearity and suitable steepness. Detailed visualization of the successive shift of reflectance response as a function of the analyte refractive index variance is illustrated in Figure 10a. The response characteristics of the sensing element with proposed parameters are: sensitivity, S=49.5 RIU^−1^; resolution limit, $2\times 10^{-4}$ RIU; and $\delta R_{p}=0.01$. The minimal step between the neighboring curves at the reference incidence angle (see the blue vertical line in the Figure 10a) is $\Delta R_{p}=0.05$.

Although comparison studies of the SPR and SPP-WG sensors showed less sensitivity in the second case ([2] and references therein), the designed structure may achieve equivalent results as a pure SPR arrangement, namely by the better figure of merit. As the typical SPR sensors are based on the response analysis represented by a plasmonic resonance minima, in presented study, we exploited properties of waveguide resonance dip modified by optical Fano effect.

It needs to be emphasized that the linearity of the sensitivity on the interval from 1.330 to 1.335 (Figure 10b) ensures reliable sensor functionality. Moreover, the concatenation of plasmonic nanostructure with waveguiding substructure leads to field intensity enhancement, enabling an improvement of resolution ability.

## Figures and Tables

**Figure 1 nanomaterials-09-01227-f001:**
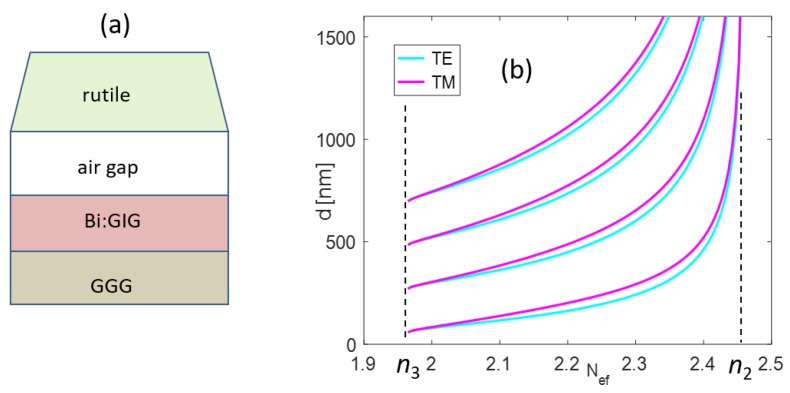
(**a**) Planar waveguide scheme; and (**b**) waveguide thickness *d* versus effective refractive index $N_{ef}$ up to third-order guided modes ($n_{2},\;n_{3}$ denote refractive indices of planar waveguide and GGG substrate, respectively).

**Figure 2 nanomaterials-09-01227-f002:**
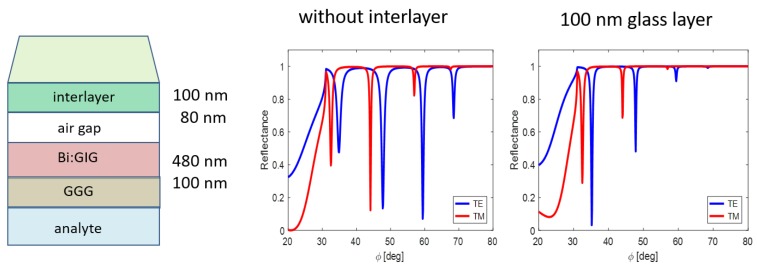
Waveguide resonance states distribution at the angular scale without glass interlayer (centre picture) and with this one.

**Figure 3 nanomaterials-09-01227-f003:**
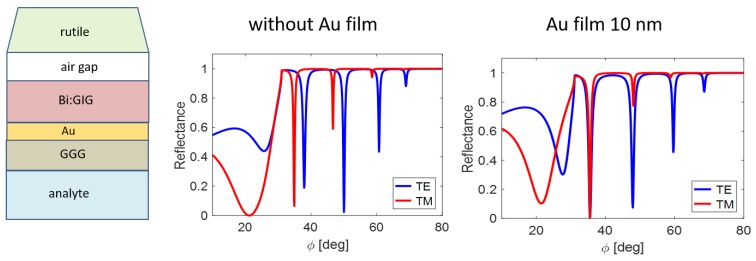
Angular shift of the TE resonance states due the Au film inserting (n=0.1838−3.4310i [18]) below the waveguide layer (air gap 100 nm, Bi:GIG 520 nm).

**Figure 4 nanomaterials-09-01227-f004:**
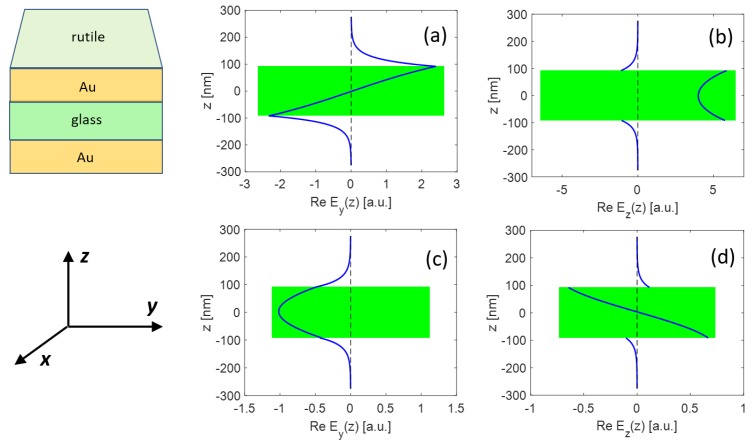
Electric field components in the MIM structure Au–SiO_2_ 184 nm–Au coupled with rutile prism. Incidence plane is perpendicular to the *x* axis. (**a**,**b**) LR SPP with odd longitudinal field symmetry at resonance incidence angle 19.8 deg; and (**c**,**d**) SR SPP with even longitudinal field $E_{y}\left(z\right)$ symmetry, incidence angle 44.4 deg.

**Figure 5 nanomaterials-09-01227-f005:**
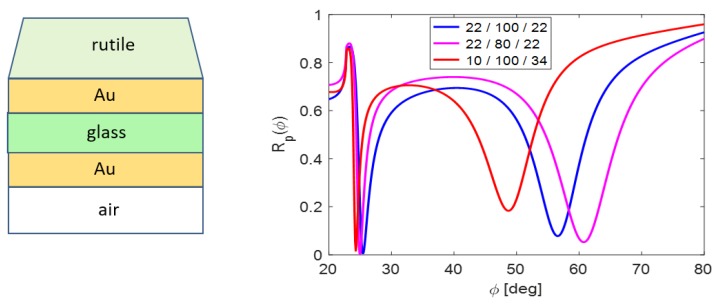
Plasmonic resonance states in the MIM structure with thin Au films: the LR SPR minima on the left (*ϕ* ≈ 25°); and the SR SPR dips on the right. The nanometer values of layer thicknesses in the legend are ordered from the closest to the prism to the air substrate.

**Figure 6 nanomaterials-09-01227-f006:**
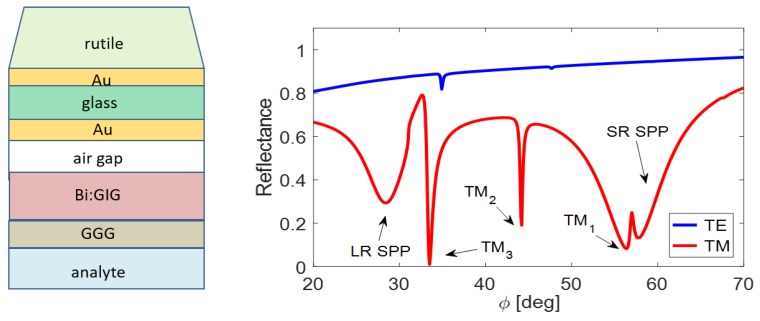
Reflectance response from plasmonic-waveguiding structure with marked resonance states. Finite layer thicknesses in nm (from top to bottom): 22/100/22/80/480/100.

**Figure 7 nanomaterials-09-01227-f007:**
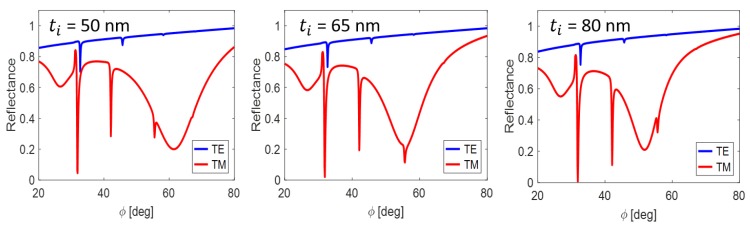
Mutual coupling of the TM_1_ waveguide mode and the short-range plasmonic mode for the layer thicknesses of the structure 10/$t_{i}$/34/80/450/100 (see previous figure for explanation).

**Figure 8 nanomaterials-09-01227-f008:**
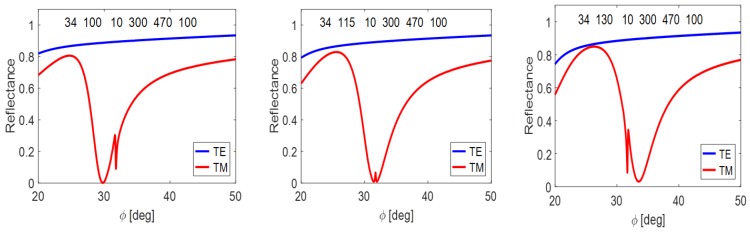
Mutual coupling of the TM_3_ waveguide mode and the long-range plasmonic mode tuned by the $t_{i}$ size. The layer proportions are specified for each structure.

**Figure 9 nanomaterials-09-01227-f009:**
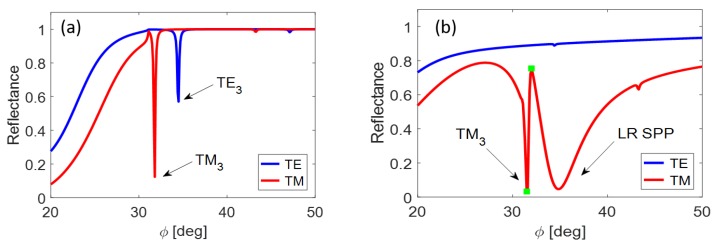
Mutual coupling of the TM_3_ waveguide mode and the long-range plasmonic mode: (**a**) layer thicknesses 0/140/0/170/470/100 (WG modes only); and (**b**) layer thicknesses 34/140/10/170/470/100 (coupled resonance states).

**Figure 10 nanomaterials-09-01227-f010:**
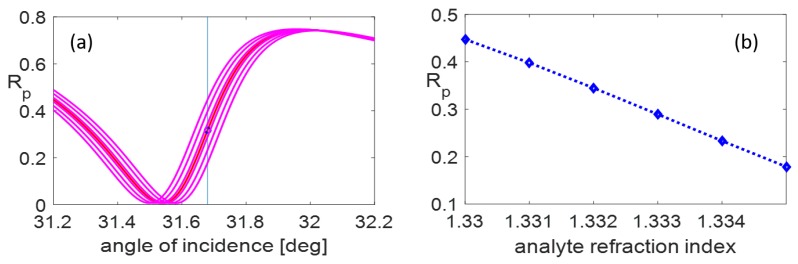
Illustrations of the sensitivity analysis: (**a**) details of reflectance curves corresponding to the scale of analyte refractive index in figure (**b**) with marked reference incidence angle; and (**b**) linearity of the $R_{p}$ vs. $n_{a}$ dependence.

**Table 1 nanomaterials-09-01227-t001:** Penetration depth of several guided modes (550 nm waveguide layer thickness) into the GGG substrate.

		Penetration Depth (nm)	
**Mode Order**	**0**	**1**	**2**
TE	36	44	94
TM	36	47	198

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
