# Peer review of "Design of Plasmonic-Waveguiding Structures for Sensor Applications"

_nanomaterials, 2019, doi:10.3390/nano9091227_

Round 1

Reviewer 1 Report

The term "medical applications" in the title is misleading since the application described belong to the general type of sensors. No specific application of medical/biological material is considered. The authors should consider a better fitted title for their work.

In section 1, the authors mention that the numerical results are obtained using rigorous coupled-waves algorithm (RCWA). More info and proper references should be given.

In Fig. 1 d, n2 and n3 should be clearly defined in the text.

Fig. 2 should be modified to illustrate the wave incidence and the angle of incidence φ.

In reference [11]: "...Plasmonic astructures with waveguiding effect..." should be "...Plasmonic structures with waveguiding effect..."

Author Response

Reviewer 1

Open Review

English language and style

( ) Extensive editing of English language and style required
( ) Moderate English changes required
(x) English language and style are fine/minor spell check required
( ) I don't feel qualified to judge about the English language and style

Yes

Can be improved

Must be improved

Not applicable

Does the introduction provide sufficient background and include all relevant references?

( )

(x)

( )

( )

Is the research design appropriate?

(x)

( )

( )

( )

Are the methods adequately described?

( )

(x)

( )

( )

Are the results clearly presented?

( )

(x)

( )

( )

Are the conclusions supported by the results?

( )

(x)

( )

( )

Comments and Suggestions for Authors

The term "medical applications" in the title is misleading since the application described belong to the general type of sensors. No specific application of medical/biological material is considered. The authors should consider a better fitted title for their work. In section 1, the authors mention that the numerical results are obtained using rigorous coupled-waves algorithm (RCWA). More info and proper references should be given. In Fig. 1 d, n2 and n3 should be clearly defined in the text. Fig. 2 should be modified to illustrate the wave incidence and the angle of incidence φ. In reference [11]: "...Plasmonic astructures with waveguiding effect..." should be "...Plasmonic structures with waveguiding effect..."

Submission Date

30 July 2019

Date of this review

05 Aug 2019 20:23:14

Answers:

The manuscript title has been modified to better agreement with the content Reference related to the RCWA is added See the first paragraph of 2.1 or the caption in Figure 1: The bismuth-doped gadolinium iron garnet waveguide layer of thickness d (Bi:GIG, n2=2.4619-0.0042i) is prepared on a gallium-gadolinium garnet substrate (GGG, n3=1.9648) See the new paragraph before Fig. 2 for explanation: The incidence angle fi is related to the prism base/structure interface; in particular, to the interface prism base/interlayer in the Figure 2. corrected

Reviewer 2 Report

I have reviewed the theoretical work done by Vlcek and co-workers titled as Design of plasmonic-waveguiding structures for medical applications. They authors proposed multiple schemes to enhance the sensitivity and performance detect desired features in bio and chemical compounds. This work has certain novelties and could be useful for readers from general plasmonics studies. The manuscript is well-written, the calculations results are convincing. I recommend this accept this work after a minor revision.

Below are some of my details suggestions before acceptance:

1) The introduction session have room to be further improved. There has intense research activities and achievements using atomic-thickness two-dimensional materials as bio and chemical detection and sensing, the progress of two-dimensional plasmonics is also making very impressive progress, it would be great if the authors could include these new developments and achievements in the introduction, so to give the readers a much broader view. For example, the following works had very good demonstrations on these developments: Science 349, 165-168 (2015); Nature 557, 530 (2018); Science Advances 3, e1701247; Nature Photonics, 4, 611-622 (2010); Nature Biomedical Engineering 3, 427-437 (2019); Nature materials 14, 1217 (2015).

2) Another important question is the suitable operating frequency range, it is unclear to me what is the best suitable frequencies for these designed structures, it would be the good of the authors could comment and address this issue.

Author Response

Reviewer 2

Open Review

English language and style

( ) Extensive editing of English language and style required
(x) Moderate English changes required
( ) English language and style are fine/minor spell check required
( ) I don't feel qualified to judge about the English language and style

Comments and Suggestions for Authors

I have reviewed the theoretical work done by Vlcek and co-workers titled as Design of plasmonic-waveguiding structures for medical applications. They authors proposed multiple schemes to enhance the sensitivity and performance detect desired features in bio and chemical compounds. This work has certain novelties and could be useful for readers from general plasmonics studies. The manuscript is well-written, the calculations results are convincing. I recommend this accept this work after a minor revision.

Below are some of my details suggestions before acceptance:

1) The introduction session have room to be further improved. There has intense research activities and achievements using atomic-thickness two-dimensional materials as bio and chemical detection and sensing, the progress of two-dimensional plasmonics is also making very impressive progress, it would be great if the authors could include these new developments and achievements in the introduction, so to give the readers a much broader view. For example, the following works had very good demonstrations on these developments: Science 349, 165-168 (2015); Nature 557, 530 (2018); Science Advances 3, e1701247; Nature Photonics, 4, 611-622 (2010); Nature Biomedical Engineering 3, 427-437 (2019); Nature materials 14, 1217 (2015).

2) Another important question is the suitable operating frequency range, it is unclear to me what is the best suitable frequencies for these designed structures, it would be the good of the authors could comment and address this issue.

Submission Date

30 July 2019

Date of this review

14 Aug 2019 16:40:04

Answers:

References to several recent related works were added in the introduction See the last paragraph in Sect. 1: The presumed experimental arrangement (detection part) for sensor response analysis is based on lock-in detection technique. In the first step of structure testing tens of kHz operating frequency range will be applied.

Reviewer 3 Report

The manuscript contains interesting design calculations on the coupling of surface plasmon polariton and optical waveguiding mode in multilayer systems for sensing applications. However, the basic principles of this approach and several examples have been published already in Ref. 11. Only the treatment of so called MIM structures containing two metal layers instead of a single one is new in the present manuscript. Furthermore, though mentioned in the title, medical applications are not addressed. In a revision, the advantage of the MIM structure in comparison to other designs should be clearly elaborated, and examples of medical applications should be addressed.

The data on the refractive index of the interlayer are a bit confusing. First, it is referred to [11], where an Al-doped zinc oxide with n = 1.8 is specified, then a glass layer with n = 1.5 is mentioned. What is correct?

There is a typing error in Ref. 3; it must read: … 2015, 48, 325303.

Author Response

Reviewer 3

Open Review

English language and style

( ) Extensive editing of English language and style required
(x) Moderate English changes required
( ) English language and style are fine/minor spell check required
( ) I don't feel qualified to judge about the English language and style

Yes

Can be improved

Must be improved

Not applicable

Does the introduction provide sufficient background and include all relevant references?

(x)

( )

( )

( )

Is the research design appropriate?

(x)

( )

( )

( )

Are the methods adequately described?

(x)

( )

( )

( )

Are the results clearly presented?

( )

( )

(x)

( )

Are the conclusions supported by the results?

( )

(x)

( )

( )

Comments and Suggestions for Authors

The manuscript contains interesting design calculations on the coupling of surface plasmon polariton and optical waveguiding mode in multilayer systems for sensing applications. However, the basic principles of this approach and several examples have been published already in Ref. 11. Only the treatment of so called MIM structures containing two metal layers instead of a single one is new in the present manuscript. Furthermore, though mentioned in the title, medical applications are not addressed. In a revision, the advantage of the MIM structure in comparison to other designs should be clearly elaborated, and examples of medical applications should be addressed. The data on the refractive index of the interlayer are a bit confusing. First, it is referred to [11], where an Al-doped zinc oxide with n = 1.8 is specified, then a glass layer with n = 1.5 is mentioned. What is correct? There is a typing error in Ref. 3; it must read: … 2015, 48, 325303.

Submission Date

30 July 2019

Date of this review: 12 Aug 2019 11:33:45

Answers

(1) In the previous (Ref. 11 in original manuscript) the magneto-optical activity of ferromagnetic garnet has been applied as the principal effect.

The concluding section has been supplied by the following paragraph:

Although comparison studies of the SPR and SPP-WG sensors showed less sensitivity in the second case (\cite{Isaacs} and references therein), the designed structure may achieve equivalent results as a pure SPR arrangement, namely by the better figure of merit. As the typical SPR sensors are based on the response analysis represented by a plasmonic resonance minima, in presented work we exploit properties of waveguide resonance dip modified by optical Fano effect.

(2) The manuscript title has been modified to better agreement with the content

(3) The MIM substructure enables among other excitation of long range SPP with large penetration depth; moreover, this SPP type coupled with a guided mode gives desired sensing quality – see the text before Fig. 5.

(4) A choose of different interlayer material is newly explained in the paragraph below the Table 1:

In the previous work the magneto-optical activity of ferromagnetic garnet has been applied as the principal effect. Therefore, only one gold layer with the Al-doped zinc oxide (n=1.8) interlayer has been used. For the SPP-PWG system discussed here an interlayer with lower refractive index is of advantage in the proposed metal-insulator-metal structure - see the Subsection 2.2.

(5) corrected

Round 2

Reviewer 1 Report

The manuscript has been revised by the authors according to the comments of the initial review and it is now suitable for publication.

Reviewer 3 Report

no further comments